# Comparing text mining and manual coding methods: Analysing interview data on quality of care in long-term care for older adults

**Coen Hacking** [1,2]*, **Hilde Verbeek**[1,2], **Jan P. H. Hamers**[1,2], **Sil Aarts**[1,2]

1 Faculty of Health Medicine and Life Sciences, Department of Health Services Research, CAPHRI Care and Public Health Research Institute, Maastricht University, Maastricht, The Netherlands, 2 The Living Lab in Ageing & Long-Term Care, Maastricht, The Netherlands

* c.hacking@maastrichtuniversity.nl

## Abstract

### Objectives

In long-term care for older adults, large amounts of text are collected relating to the quality of care, such as transcribed interviews. Researchers currently analyze textual data manually to gain insights, which is a time-consuming process. Text mining could provide a solution, as this methodology can be used to analyze large amounts of text automatically. This study aims to compare text mining to manual coding with regard to sentiment analysis and thematic content analysis.

### Methods

Data were collected from interviews with residents (n = 21), family members (n = 20), and care professionals (n = 20). Text mining models were developed and compared to the manual approach. The results of the manual and text mining approaches were evaluated based on three criteria: accuracy, consistency, and expert feedback. Accuracy assessed the similarity between the two approaches, while consistency determined whether each individual approach found the same themes in similar text segments. Expert feedback served as a representation of the perceived correctness of the text mining approach.

### Results

An accuracy analysis revealed that more than 80% of the text segments were assigned the same themes and sentiment using both text mining and manual approaches. Interviews coded with text mining demonstrated higher consistency compared to those coded manually. Expert feedback identified certain limitations in both the text mining and manual approaches.

### Conclusions and implications

While these analyses highlighted the current limitations of text mining, they also exposed certain inconsistencies in manual analysis. This information suggests that text mining has

**Data Availability Statement:** The code is now available on Zenodo: https://zenodo.org/doi/10.5281/zenodo.8391746. Our interview data will not be publicly available due to the privacy of our

participants. Upon request, our interview data may be provided with restrictions. Data are available from the Living Lab in Ageing and Long-Term Care (contact via Sil Aarts (ouderenzorg@maastrichtuniversity.nl) for researchers who meet the criteria for access to confidential data.

**Funding:** The author(s) received no specific funding for this work.

**Competing interests:** The authors have declared that no competing interests exist.

the potential to be an effective and efficient tool for analysing large volumes of textual data in the context of long-term care for older adults.

## Introduction

In recent years, client perspectives have become increasingly important in long-term care (LTC) for older adults when assessing the quality of care [1–3]. To gain insight into these perspectives, textual data are often collected, such as electronic health records, policy documents or transcribed interviews with various stakeholders, including residents of nursing homes [2,4]. When interviews are conducted with stakeholders in nursing homes, textual data may be collected by transcribing audio recordings verbatim from interviews (i.e. literally translating voice into text), and are often referred to as transcripts. This type of data collection often results in large amounts of textual data. To be able to analyse these data, researchers often conduct a so-called coding analysis, which involves manually analysing each transcript (stemming from an interview) to identify text fragments that are relevant to the objective at hand (often a research question) [2,5]. Each key fragment is summarised using codes (i.e. summaries of several words) that reflect the condensed meaning of that specific fragment [5]. The codes are then clustered based on their similarity, and are grouped into themes [5]. These themes convey a certain topic which is of relevance to the transcript at hand, which often provides a direct or indirect answer to the research question [5]. Although this type of coding is typically performed in a bottom-up manner, it is also possible to apply a top-down approach, in which case a set of themes is constructed in advance [3]. Since text analysis through coding is known to be very time-consuming and prone to bias due to the subjectivity of the researchers, coding is often performed independently by two or more researchers, thereby ensuring a certain level of objectivity. Manual analysis is never completely objective, as researchers are prone to human biases such as generalisations, inferences, and interpretations [6,7]. which compromise the reproducibility and limit the amount of data that can be analysed.

To overcome the aforementioned drawbacks, text mining could offer a possible solution. Text mining is the process of transforming unstructured text into structured data in order to gain new information and knowledge [8]. and has already been used for knowledge discovery in other domains of health care [4,9–13]. Knowledge discovery is the process of extracting useful information from a collection of data; for example, a study conducted on electronic health records discussed how text mining could be used to group pathology reports and discharge summaries, based on similar word occurrences [10]. Another study that focused on organising clinical narratives concluded that text mining could be applied to clinical narratives to identify keywords that could help in classifying physiotherapy treatments [4]. These examples highlight the usefulness of text mining in the health care domain.

Recent advancements in the field of text mining have ushered in a variety of new techniques, each with its unique focus and application [14–19]. Some models are particularly good at generating context-aware, human-like text, while others excel at incorporating multi-modal data, such as text and images, for a more comprehensive analysis [14–16]. Moreover, there is a growing emphasis on adapting these models to run efficiently on consumer-grade hardware [17]. Despite these strides in technology, there are still significant challenges in achieving the level of accuracy required for some tasks, and in many cases, human expertise continues to outperform automated methods [17].

To understand the potential usefulness of text mining for qualitative research in long-term care for older adults, it should be compared to the current gold standard of manual coding

[20]. This study aims to compare a text mining approach to a manual approach in terms of accuracy, consistency, and expert feedback. Accuracy is a measure of the degree to which the results from the text mining approach are similar to those of the manual approach, whereas consistency is defined as the degree to which an approach (i.e. text mining or manual) finds the same themes for similar pieces of text. Expert feedback is collected to show whether the analyses conducted through text mining are perceived to be correct.

## Materials and methods

### Study design

In this study, a comparison was conducted between the use of manual and text mining approaches in a sentiment analysis and a thematic content analysis of qualitative data accumulated in an LTC setting. Two different text mining models were constructed: (i) a sentiment analysis model, and (ii) a thematic content analysis model [21,22]. Each model was then compared to the respective manual coding approach, based on an accuracy evaluation, a consistency evaluation and expert feedback.

### Sample and participants

Data were collected as part of a project entitled 'Connecting Conversations', which aimed to assess the experienced quality of care in nursing homes from different perspectives [2,23]. This was achieved by interviewing residents, family members and care professionals at different nursing homes in the South of Limburg [2,23].

A total of n = 250 interviews were conducted at five different LTC organizations in the southern part of the Netherlands. From those interviews, 234 were transcribed (16 could not be transcribed due to poor audio quality). From the remaining 234 interviews, 61 were analysed manually using thematic content analysis. In addition, 103 interviews were analysed manually using sentiment analysis. All analysis in the manuscript were performed using those 61 and 103 interviews for the thematic content analysis and sentiment analysis respectively.

All interviews were conducted between January 2018 and December 2019. A diverse set of wards were included, including those for older people with dementia [23]. A total of n = 35 interviewers conducted the interviews. These interviewers were part of the project 'Connecting Conversations,' which aims to assess the experienced quality of care in nursing homes from the resident's perspective. They primarily come from a long-term care setting and have received specialized training to conduct these interviews. For a more comprehensive understanding of the 'Connecting Conversations' project, see Sion et. al. 2020a [2]. The medical ethical committee of Zuyderland (the Netherland) approved the study protocol (17-N-86). Information about the study was provided to all interviewers, residents, family members and caregivers by an information letter. All participants provided written informed consent: residents with legal representatives gave informed consent themselves (as well as their legal representatives) before and during the conversations.

### Data

The interviews were anonymously collected in the form of audio recordings and were transcribed verbatim (in Dutch) [2]. Personally identifiable information was removed from the transcripts before being coded. The data were coded by three research experts, each working in the *Living Lab on Ageing and Long-Term Care* for over 5 years. All these experts have a minimum of ten years of experience in conducting qualitative research. A total of 103 transcripts were manually coded regarding the sentiment [24]. In this analysis, text segments were

manually coded as being either 'positive' or 'negative'. However, text segments were only coded if the text discussed a topic relevant to the nursing home. A total of 61 transcripts were manually coded using INDEXQUAL, a thematic framework for defining the quality of LTC [3]. The themes provided by INDEXQUAL are 'context', 'nursing home', 'person', 'expectations', 'personal needs', 'past experiences', 'word of mouth', 'experiences', 'care environment', 'relationship-centred care', 'experienced quality of care', 'perceived care services', 'perceived care outcomes' and 'satisfaction' [2,3]. In both cases, transcripts were coded using MAXQDA, and these codes were exported to develop a text mining approach [25].

## Text mining models

The models presented in the current study were created using deep learning, a method in which artificial neural networks (ANNs) are used to learn automatically from input data [26]. A Dutch base language model called RobBERT was used [22]. The advantage of using such a model is that language knowledge can be learned from a large dataset of arbitrary (Dutch) text. Two models were developed in the current study: a sentiment analysis model, and thematic content analysis model. The code for the models can be found at: https://doi.org/10.5281/zenodo.8391747.

**Sentiment analysis.** Sentiment analysis is the process of computationally identifying the sentiment expressed in a piece of text [8,27]. For example, the sentence 'It's a good day' could be identified as being positive, while the sentence 'It's a bad day' could be identified as being negative. The sentence 'Today I went for a walk,' could be neutral, as it does not convey whether the walk was experienced as a positive or negative event. Coded text segments were passed directly as input to the model, without modification. The sentiment analysis model was trained to classify the sentiment of a given piece of text into one of two categories, i.e. positive or negative. A positive or negative code was only assigned when it was perceived as being relevant to improving the quality of care [24].

**Thematic content analysis.** As part of the thematic content analysis, the model was trained to identify the themes present in each piece of text and to classify them into the relevant themes of the INDEXQUAL coding scheme. Since the number of coded text segments (n = 3867) was insufficient to allow the model to learn all the themes and sub-themes (n = 16), only the main themes were used: 'Experienced quality of care', 'Experiences', 'Expectations' and 'Context' [3]. Each code containing a sub-theme was changed to one of these main themes, and the model was designed to be able to identify multiple themes that may be present in a text segment.

## Evaluation

The text mining models were analysed in three ways: an accuracy evaluation, a consistency evaluation, and using expert feedback. The accuracy analysis assessed the ability of each model to correctly classify or predict outcomes based on the input data, while the consistency analysis evaluated their ability to produce consistent results over multiple runs or when applied to different datasets, and expert feedback was used to provide additional insight into the performance and potential biases of the models [28–30].

**Accuracy.** The accuracy evaluation aimed to calculate the percentage of text segments that were assigned the same codes in both the text mining approach and the manual approach [8,27,31]. For example, if the text mining model for sentiment analysis assigned the same sentiment as the manual approach for all of the sentences, then the model would be considered 100% accurate. To calculate the accuracy, training and validation sets were used: the training set was used to provide feedback to the model to help improve it (i.e. supervised learning),

while the validation set was used to evaluate whether what the model had learned so far could be generalised to data that it had not had the chance to learn from [28]. The total amount of data was split, with 90% forming the training set and 10% the validation set. The accuracy score from the validation set was reported, as this is more representative of how a model would perform on unseen data [28]. A confusion matrix was used to display the results of the accuracy evaluation. Such a matrix shows the different cases for each possible choice that either the manual or text mining approach can make. Accuracy was calculated using the formula: (TP + TN) / (TP + TN + FP + FN). In this case, TP is the true positive (i.e. where a code is present in both analyses), TN is the true negative (i.e. where a code is absent in both analyses), and FP is the false positive (i.e. where a code is predicted to be present but is absent in the manual analysis), while FN is the false negative (i.e. where a code is predicted to be absent but is present in the manual analysis). These components help us assess the accuracy of the model's predictions and its performance overall [28].

**Consistency.** In the consistency evaluation, both the manual and text mining approach were analysed to determine the consistency of each approach individually. When a coded text is consistent, the expected outcome is that each sentence that is semantically similar will be coded in the same way. A consistency evaluation was conducted by comparing the assigned themes or sentiment between similar sentences; for example, if two sentences were semantically very similar, then it would be expected that these sentences would also be coded with the same themes, and if two sentences were semantically very different, it would be less likely that these would be coded in the same way [30,32].

**Expert feedback.** To determine whether the output of the models was reliable and comparable to that of manual coding, feedback was collected from the original research experts. This information was collected from three of the research experts who coded the original data, all of whom worked at the Living Lab on Ageing and Long-Term Care for over 5 years. All their feedback was captured in an audio-recorded interview.

The research experts were shown three coded transcripts and were asked to give feedback on them. Without their knowing, the research experts were shown one transcript that had been left unmodified manually coded transcripts (i.e. a transcript that contained the codes as previously analysed by the research experts themselves). After being shown each individual transcript, the research experts were asked to provide feedback on that transcript overall. Their feedback was then analysed to discover potential issues with the text mining approach.

Following this, the research experts were given one large transcript from the validation set in which they were shown both the manual and text mining versions next to each other. This type of comparison allowed them to comment on why the differences between the approaches arose. Their feedback was also used to highlight issues with the accuracy analysis.

## Results

### Accuracy

**Sentiment analysis.** The results show that the overall accuracy for the sentiment between the manual approach and the model was 81.8%. Fig 1 displays the results of the sentiment analysis in the form of a confusion matrix. It can be seen from the figure that most of the text in the transcripts was not coded with a sentiment, either through the manual process or through text mining. Manually coded text with a negative sentiment was only recognised as positive by text mining in 0.1% of cases, and only 0.3% of the text that was manually coded with a positive sentiment was recognised by text mining as negative. The average accuracy over all transcripts was 88.7% with standard deviation of 8.6%. The minimum was accuracy was 52.1% and the maximum accuracy was 99.6%.

**Text mining**

**Fig 1. Confusion matrix comparing sentiment analysis results of the manual and text mining approach.** The matrix compares manual coding (rows) against text mining predictions (columns) for sentiment values of the text. Each cell within the matrix represents the percentage occurrence of a particular sentiment alignment (or misalignment) between the manual and text mining approaches. The y-axis of each matrix represents the sentiment as determined through manual analysis, while the x-axis indicates the text mining predictions. The diagonal cells (from top left to bottom right) illustrate the percentage of agreement between the two methods, whereas all off-diagonal cells indicate discrepancies. For instance, the cell at the intersection of the "Positive" row and the "Negative" column displays instances where text was manually coded as positive but was predicted as negative by text mining.

**Thematic content analysis.** A comparison was conducted between the manually coded INDEXQUAL themes and the codes predicted by the model, and the results indicated that the model achieved an accuracy of 83.7%. Fig 2 shows the confusion matrices for the validation set. For all of the themes in general, it was found that most of the text segments that weren't coded by the manual approach, were also not coded by the text mining approach. For the theme 'Context', we found that the text mining approach assigned a code to a text segment much more often compared to the manual approach. The themes of 'Context' and 'Expectations' were absent from most of the manually coded text (in 87.9% and 95.2% of cases, respectively). The themes of 'Experienced Quality of Care' and 'Experiences' were identified correctly by the text mining approach in a higher percentage of text segments compared to 'Context' and 'Expectations'; however, 'Experienced Quality of Care' and 'Experiences' also had higher rates of false positives and false negatives. False positives were cases where text mining incorrectly assigned a particular theme to text segment, and false negatives were cases where text mining incorrectly failed to assign a theme. "The average accuracy over all transcripts was 81.9% with a standard deviation of 8.5%. The minimum accuracy of any transcript was 43.1% and the maximum was 93.4%.

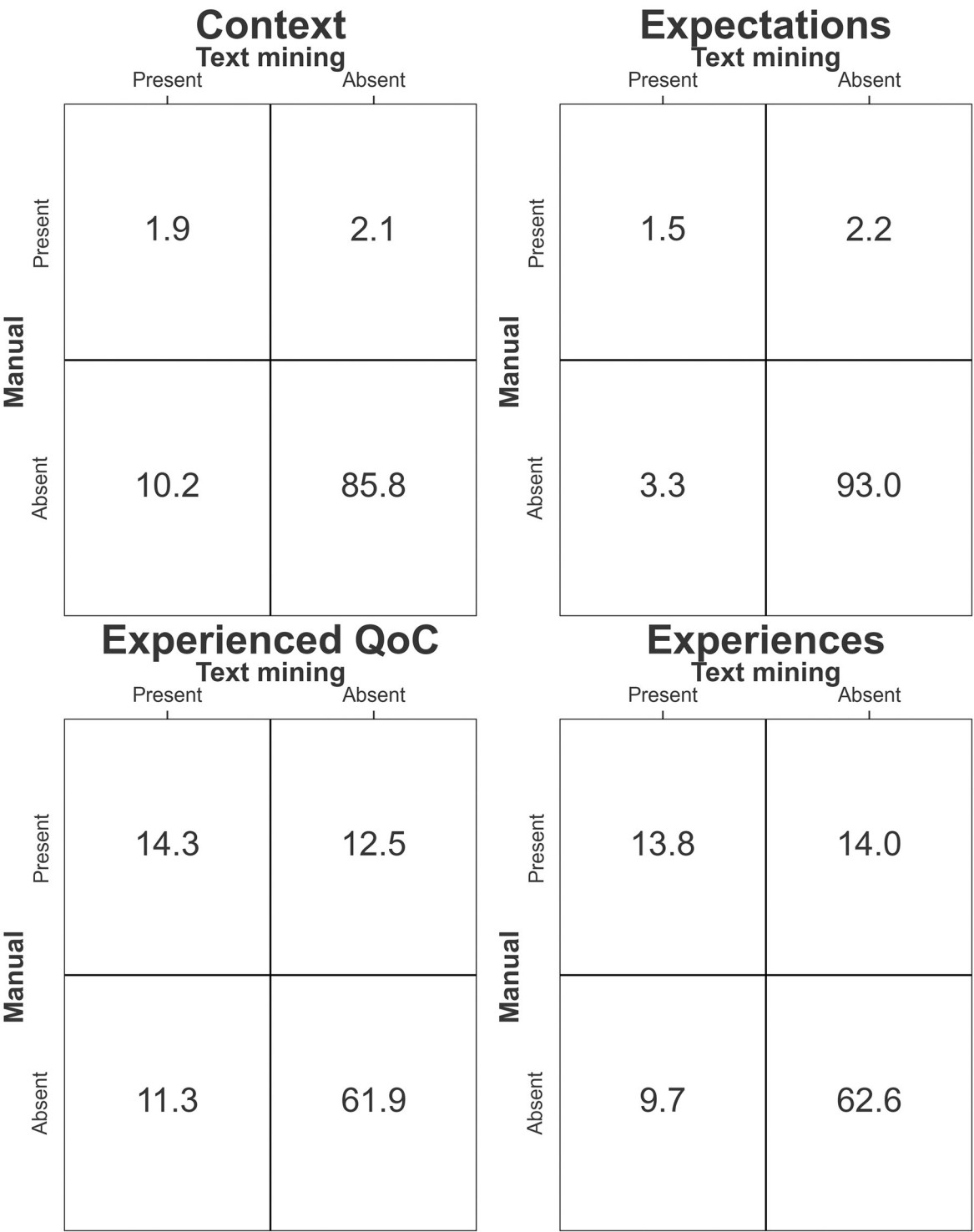

**Fig 2. Comparison of results from the thematic content analysis.** A confusion matrix is shown for each of the main INDEXQUAL themes (Experienced quality of care, Experiences, Expectations and Context). The y-axis of each matrix represents the presence or absence of a theme as determined through manual analysis, while the x-axis indicates the text mining predictions. Cells on the diagonals capture instances of agreement between manual coding and text mining for each theme. Off-diagonal cells detail discrepancies, indicating false positives or false negatives. Percentages within cells show the proportion of occurrences for each scenario in relation to the total dataset.

**Table 1. Overview of the consistency of the manual and text mining approaches in regarding the sentiment analysis.**

| Theme | Manual (%) | Text mining (%) |
|---|---|---|
| Positive | 68.3 | 74.4 |
| Negative | 67.6 | 73.8 |

## Consistency

**Sentiment analysis.** Consistency scores were calculated as part of the sentiment analysis, as shown in Table 1. Semantic similarity is a value between 0% and 100%, where a higher percentage indicates that the results were more consistent [33]. On average, the transcripts coded using the sentiment analysis model were more consistent than those coded using the manual approach.

**Thematic content analysis.** As is shown in Table 2, the text mining approach was more consistent when coding sentences related to experienced QoC and experiences. These were also the themes that occur most often in the interviews. On average the text mining approach was more consistent using the current metric. While, the results displayed a low consistency, it should be noted that only limited context was taken into account. This increased the perceived similarity of sentences and therefore decreases the consistency.

## Expert feedback

Overall, the research experts expressed a mixed-to-positive assessment of the analysis of the transcripts. While they were most positive about the manually coded transcript, they were unable to distinguish it from the transcript coded by the text mining algorithm in the training set. In contrast, the text mining approach in the validation set was recognized by the research experts as having a lower level of accuracy (e.g., smaller coded text segments compared to the manual codes). The research experts identified certain themes, such as "Context" and "Expectations," as posing greater difficulties for the algorithm, whereas other themes, such as "Experienced Quality of Care" and "Experiences," were coded more similarly by both the algorithm and the research experts. The experts acknowledged that coding was generally a challenging task.

> "I don't find the coding to be poor. I notice that the codes about which the text mining approach is wrong, we've also had deliberations."

> "[Text mining] isn't not all perfect, however it does allow us to analyse much more interviews."

The research experts were presented with a transcript from the validation set, where both the manual and text mining versions were presented side by side to enable the research experts to explain the differences between the approaches. Most of the feedback from the research

**Table 2. Overview of the consistency of the manual and text mining approaches regarding various themes.**

| Theme | Manual (%) | Text mining (%) |
|---|---|---|
| Experienced QoC | 51.8 | 58.9 |
| Experiences | 54.0 | 59.1 |
| Expectations | 59.5 | 61.8 |
| Context | 59.4 | 62.2 |
| Average | 56.2 | 60.5 |

experts focused on codes that were similar between the two approaches or where the text mining approach incorrectly coded something. However, according to the experts, some codes were coded correctly by the text mining approach, but not by the manual approach.

*"Yes, we've missed that one, seems logical to me."*

*"Yes, [similar to the other] we missed that one as well."*

Although the instances of text mining finding errors in the manual codes were few, they negatively impacted the accuracy analysis. This is because such codes were regarded as false positives. Additionally, there was at least one instance where the text mining algorithm had coded the same information at a different location in the text.

*"Here, the model applied the theme of quality of care [instead of where we coded it]."*

## Discussion

This study compared two approaches to coding text, a text mining approach and a manual approach, and carried out two types of analysis: a sentiment analysis and a thematic content analysis. The two approaches were compared in terms of their accuracy and consistency, and based on expert feedback. The results showed that for most text segments, the approaches were coded in a similar fashion. However, further analyses also showed that there were key differences in coding between the text mining approach and the manual approach in terms of accuracy and consistency.

The results of an accuracy analysis showed that the text mining models coded text with the same themes as the manual approach in more than 80% of cases. However, it was found that the number of false positives and false negatives were relatively high compared to the true positives. This indicates that the actual similarity (i.e. for text containing more coded segments) between the methods may be lower. One of reasons for the discrepancies between the manual and text mining approaches is that many manually coded text segments contain more than one theme; for example, 19% of all of the text coded with the theme 'Experiences' was also coded by the research experts with other themes, such as 'Experienced quality of care' or 'Expectations'. The presence of overlapping themes in text can pose a challenge for text mining models, as this makes it more difficult to accurately determine which text characteristics correspond to each theme. In addition, the complexity and variability of natural language and the current limitations of text mining algorithms may also contribute to the lower accuracy of text mining models [34–37]. The variance of the accuracy between transcripts shows that a possible reason for lower accuracies could be due to factors that vary between transcripts, such as the quality of the transcription, the nature of the language used by the participants, or contextual factors that were not taken into account by the text mining or manual approach.

The results of a consistency analysis suggested that the current text mining models were able to produce more consistent codes for semantically similar sentences across all interviews compared to the manual analyses. However, the measured difference in consistency between the approaches was less than 5% on average. This could be explained by the fact that the text mining approach learned from the manual codes, and hence the text mining models also exhibited the same type of inconsistencies to a certain degree [38,39].

Feedback from the research experts suggested that text mining could be a valuable supplement to traditional qualitative analysis methods, and could provide a more efficient and

objective way of analysing large amounts of text data [40,41]. However, research experts were able to identify flaws in both methods of analysis. This could be because research experts had more knowledge about the subject of the analyses and could therefore recognise wider patterns [42,43]. However, it was difficult for human experts to distinguish between the codes they had assigned manually and codes that were assigned by the text mining model. When the experts were able to compare the codes created by the text mining approach and their own manual codes, they reported that they had also missed certain text segments when they originally coded the interviews. These segments were discovered and coded by the text mining models. This finding suggests that text mining models could be helpful for manual analysis, as demonstrated using recent methods such as InstructGPT and MM-CoT [14–16]. These methods show that language models can aid in a variety of tasks, from writing cover letters to creating SPSS or Python scripts. However, these language models require human guidance to achieve the best results, as many of these tasks may be subject to human bias [38].

Using deep learning models, such as those highlighted in this study, offers a distinct advantage in terms of speed. While deep learning models can process and analyse data within seconds, manual analysis, depending on the complexity and volume of the data, can span weeks to even months [44]. However, it's essential to recognize that the results from deep learning models might not always align perfectly with those of manual analysis. As such, researchers might find the need to fine-tune the outputs generated by text mining models. Despite this, the integration of deep learning significantly accelerates the qualitative analysis process, offering a more efficient alternative to traditional methods.

Some methodological limitations must be acknowledged. Firstly, in large parts of the interviews, no codes were identified by either the text mining or manual approach. As a result, the average accuracy of the text mining models was higher than it might have been if the text were coded with a higher density. Secondly, it is important to consider the limitations of the algorithm used to calculate the sentence similarity. This algorithm has an accuracy that is limited to 66% for the classification of similar sentences [30]. This is also challenging, as it is therefore difficult to define which properties of a text segment are important in terms of the semantic similarity. For example, given four sentences regarding a resident, a nurse, a resident's family member, and a visiting doctor, it is possible to split them based on whether a person is a healthcare professional or not; however, it is also possible to split them based on whether a person is part of the nursing home staff or an outsider. Which property is more important to the similarity depends on factors such as the research question, and determining the similarity becomes more difficult with complex sentences. Moreover, it is important to consider the potential for human bias in qualitative analysis. Bias can arise from a variety of sources, including the research expert's own preconceptions and assumptions, the sampling and recruitment of participants, and the methods and techniques used to collect and analyse data [36,37]. As the text mining model learns from inherently subjective data, it also learns to apply codes with the biases that exist in the data. While the expert feedback showed that few of these cases existed, such cases can negatively impact the evaluated accuracy of text mining models. Lastly, the analysis conducted in the current study only had context window of 512 words at most, which represents a technical limitation of the method [21,22]. This limits the textual context that the models have access to. These issues can be mitigated using large language models that are better able to capture the nuances and complexities of natural language (e.g. GPT-3) [25,37,45]. Such models can also handle a larger context of words. Whereas RobBERT has a maximum context length of 512, GPT-3 has a context of 4,096. However, such large language models cannot be used on most personal computers, as they require specialised hardware to run efficiently (i.e. GPUs or TPUs with large amounts of memory) [46]. Using these via online (cloud) systems could give rise to issues regarding the privacy of the interview participants. However, recent advances have shown

that 'smaller' (i.e. more efficient) large language models can achieve similar results, and these models can be used on personal computers, unlike GPT-3 [19,47].

### Future work

Future research could focus on applying a hybrid approach that combines the text mining and manual methods. Using this approach, a text mining algorithm could be used to pre-process the text data and identify potential themes and patterns, which could then be reviewed and refined by human experts. This would allow for an efficient and objective analysis of large datasets, while also allowing for the expertise and knowledge of human experts to be incorporated. Future research should investigate whether this approach could help to reduce the potential for bias and improve the accuracy of the results.

Future work could compare multiple novel text mining models such as GPT-4 and LLaMA to show whether larger models can generate results that are better with respect to the context and more similar to the manual analysis. Comparing different models side-by-side could offer a useful way to visualize the main features and capabilities of each model, and could also facilitate the identification of any common weaknesses or limitations that may exist across some or all of the models being investigated. This could also enable the identification of areas where specific models may excel relative to others.

## Conclusions

The current study shows that text mining can be an effective tool for quickly and accurately identifying sentiment and thematic content from large amounts of textual data. Text mining can help to reduce the amount of time and resources needed to analyse textual data, making it a valuable tool for analysing large amounts of qualitative data. However, as shown in the current study, text mining has certain limitations regarding language understanding; in its current state, text mining is no substitute for manual coding, but can be seen as a helpful addition.

## Acknowledgments

The authors would like to thank the Data Science Research Infrastructure (DSRI) at Maastricht University. Without the usage of their computational resources, the current study could not have been conducted. Moreover, thanks to the contributors of the "huggingface transformers" library, as the library provided many of the components for developing the ASR model in the current study. Lastly, a special thanks to Katya Sion, Audrey Beaulen and Erica de Vries, the research experts who manually coded the transcripts and gave their feedback.

## Author Contributions

**Conceptualization:** Coen Hacking, Hilde Verbeek, Jan P. H. Hamers, Sil Aarts.

**Formal analysis:** Coen Hacking.

**Methodology:** Coen Hacking, Sil Aarts.

**Project administration:** Sil Aarts.

**Software:** Coen Hacking.

**Supervision:** Hilde Verbeek, Jan P. H. Hamers, Sil Aarts.

**Writing – original draft:** Coen Hacking.

**Writing – review & editing:** Coen Hacking, Hilde Verbeek, Jan P. H. Hamers, Sil Aarts.

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
