## [Decision Letter · Decision Letter 0]

26 Jul 2023

PONE-D-23-11158A comparison of text mining and manual coding methods in long-term care for older adults regarding quality of carePLOS ONE

Dear Dr. Hacking,

Thank you for submitting your manuscript to PLOS ONE and for your patience in awaiting the reviewers' response. Both reviewers noted the importance of our wok but after careful consideration, we feel that whilst your manuscript has merit, it does not fully meet PLOS ONE’s publication criteria as it currently stands. I shared the reviewers' view on the applied relevance of your work but also felt that methodological information, background/references about the methodological approaches and analyses were limited and at times confusing, thus impacting the reproducibility of your work. Therefore, we invite you to submit a revised version of the manuscript that addresses the points raised during the review process.

Please ensure that your decision is justified on PLOS ONE’s publication criteria and not, for example, on novelty or perceived impact. For a potential acceptance of your submission, we expect you to address all concerns raised by the two reviewers that are possible based on the data you have collected for this submission. For concerns where you would require additional data collection but are not in a position to do so, please consider their points in your response and potentially in the limitation section of your manuscript. 

We look forward to receiving your revised manuscript.

Kind regards,

Corinne Jola

Academic Editor

PLOS ONE

Journal Requirements:

Reviewers' comments:

Reviewer's Responses to Questions

**Comments to the Author**

1. Is the manuscript technically sound, and do the data support the conclusions?

Reviewer #1: Yes

Reviewer #2: Yes

2. Has the statistical analysis been performed appropriately and rigorously? 

Reviewer #1: I Don't Know

Reviewer #2: Yes

3. Have the authors made all data underlying the findings in their manuscript fully available?

Reviewer #1: No

Reviewer #2: No

4. Is the manuscript presented in an intelligible fashion and written in standard English?

Reviewer #1: Yes

Reviewer #2: Yes

5. Review Comments to the Author

Reviewer #1: Dear Authors,

Thank you for this possibility to peer review your interesting and important study paper. The study aims to understand the potential usefulness of text mining for qualitative research in long-term care for older adults. It compares text mining approach to manual approach regarding sentiment analysis and thematic content analysis in terms of accuracy, consistency, and expert feedback. This study is meaningful for the reasons which the authors state in their paper - manual analysis of large free-text-data-sets is time consuming and has risk of objectivity bias. Thus new effective analysis methods are needed.

These peer review comments are given from the point of view of non-native English user and nursing scientist who has used text-mining method in own research.

1. Is the manuscript technically sound, and do the data support the conclusions?

-Line 1: The tile might be stronger if the "interview data" would be stated in the title

-Line 38: In the abstract conclusions, I am not sure if you can say, based on your results, that text mining is potential for large data sets, as your sample was just couple of hundreds interview texts. I would prefer to conclude, that text mining is potential for analysis of free text data in the context of LTC for older adults. For future work you might suggest to test the method in large data sets (e.g. thousands-tens of thousands).

-The study presents results of original research and the sections are constructed according to journal's guidelines and are named and organized in logical order.

-The introduction section describes the judgement for the need of this study and states the aim of the study.

-Line 75: There should be reference for the gold standard of manual coding.

-Materials and Methods paragraph: You could add here your study design.

-Lines 85-86: References for the two Text mining models should be presented.

- Line 93 in Sample and Participants paragraph: A total of n=250 is confusing as earlier was said different n-numbers. In addition in Line 105 and Line 108 -> are presented different numbers. The numbers in different places needs more clarification for transparency.

Should you describe also the experts' interviews in this section?

-Line 94: Who were interviewing the participants? How many interviewers?

2. Has the statistical analysis been performed appropriately and rigorously?

-Line 113: Is the MAXQDA coding system free of charge or commercial software? Can you describe this?

-Line 121-123: Move this sentence to limitation section in discussion " Both models were capable of identifying where and how to code a text segment using a context of 512 words at most, which represents a technical limitation of the method [17,18]"

-Line 125: Who coded the sentiment analysis? One, two or more researchers?

3. Have the authors made all data underlying the findings in their manuscript fully available?

-Line 160: Could you present sample table of accuracy scores and confusion matrix?

-Line 169-171: What was semantics consistency scale between very similar - very different?

-Line 176-177: Research experts interview were not described in methods section.

-Line 199-202: The sentence is unclear: "The text mining model was less accurate in determining what may be relevant to the organisation of a nursing home, as text which was not coded manually was often coded by text mining as either negative (4.3%) or positive (5.2%)."

-Line 278: Discussion: The text is logical and clear in discussion, however I would suggest to add comparison of your results with previous studies. You have some references presenter, but not the comparison or mirroring your own results towards other studies.

-Line 328...: Could you present some health care related example instead of the phrase ".... For example, given four sentences regarding a cat, a dog, a lion, and a wolf, it is ...."

-Line 359 in Conclusions section: Here also you use wording "large data sets" ... prefereably should not overestimate suitability for large data sets based on your results, just suitability for free text analysis in this context.

-References: Majority of the references are of high quality and from the last five years. However, 1/4 of the references are older than 10 years and the peer-review of the reference is missing or unclear among the following references: Lines: 393, 400, 412, 417, 419, 428, 435, 447, 456, 459, 463, 465, 468, 472 and 475.

The references with "arXiv preprint arXiv" are submissions to a computer scientist database, but might not be yet peer reviewed or might have been even rejected from publication after submission. You should try to find the peer reviewed version of these articles and write references according to accepted/ published versions of the papers.

-Figure 1 and Figure 2: These figures do not open up for me. These needs revisions and clarifications. At the moment it looks to me, that Text mining and Manual analysis had exactly the same statistical values. I do not know how to read the Figures. In addition, you should understand the figures as stand-alone, without reading the manuscript text.

4. Is the manuscript presented in an intelligible fashion and written in standard English? PLOS ONE does not copyedit accepted manuscripts, so the language in submitted articles must be clear, correct, and unambiguous. Any typographical or grammatical errors should be corrected at revision, so please note any specific errors here.

-As I am not native English user, I do not have comments for the English grammatic or typography.

I hope these comments help you to further develop your paper and to get it even stronger.

This paper deserves to get published, but needs some more details for transparency and in order to be able to use your method for replication.

Reviewer #2: Summary Statement: This reviewer thanks the authors for their submission. Indeed, evaluation of information from clients in long-term care is relevant and methods to address their quality of care experience. More specifically, authors aim to evaluate if text mining can be accurate, consistent, and similar to expert review in both sentiment analysis and thematic content. The manuscripts is straightforward and quite readable but could be improved with further methodological and quantitative depth added as well as specific details in the discussion about how to improve the next study in the domain.

Strengths where no changes are required:

1) The methods of inclusion are well-described and include a robust number of participants (n=250) with written informed consent from 5 sites which seems adequate and appropriate for the evaluation.

2) The methods of themes into (14) key areas seemed appropriate and were well-described.

3) Evaluation of the text mining was performed using accuracy, consistency, and expert review which seemed appropriate.

Weaknesses and areas of the manuscript that could be improved through further efforts:

1) The types of text mining could be further scientifically described in the introduction that are used later in the methods section:

Authors could better represent the current challenges with coding of qualitative data with clear description of methods and their performance characteristics. Recommend add 2-3 references in the introduction that specifically outline challenges in current methods with more specificity. Other authors have described with greater detail the text mining methods such as those described in the following (or could be alternative): Pranita Mahajan, Dipti P. Rana; International Journal of Innovative Technology and Exploring Engineering (IJITEE), ISSN: 2278-3075, Volume-9 Issue-2S, December 2019; or Annu Rev Biomed Data Sci. 2021 Jul 20;4:165-187. doi: 10.1146/annurev-biodatasci-030421-030931. Epub 2021 May 26. Additionally, further details about text mining models including InstructGPT and MM-CoT would be appropriate to cover in 1 sentence in the background/introduction.

2) While the authors used a Dutch language inclusive model, only a limited number of text word concepts were available (n=512 words). While this may not require a modification, authors in the discussion should further elaborate on the implications to the findings.

3) Expert feedback is inadequately described and should include a sentence referencing exactly in what capacity the individuals are considered expert. The statistical methods utilized in the comparison should also be briefly presented in the methods.

4) Authors describe the difficulties inherent in the analyses with discrepancies which can also be due to as well the multiple themes that are present but could expand upon how this can be mitigated in the discussion.

5) Overall, the manuscript could be improved through the additional enrichment of quantitative findings as well as depth in the methodological approach.

Minor Editing Recommendations:

1) Correction recommendation: In the introduction, there is a period that needs replaced by a comma, line 48:

To be able to analyze these data, 48 researchers often conduct a so-called coding analysis [2,5]. which involves manually.

6. PLOS authors have the option to publish the peer review history of their article (what does this mean?). If published, this will include your full peer review and any attached files.

Reviewer #1: No

Reviewer #2: No

---

## [Author Response · Author response to Decision Letter 0]

9 Sep 2023

We’ve checked and updated the styling accordingly.

2. We note that you have indicated that data from this study are available upon request. PLOS only allows data to be available upon request if there are legal or ethical restrictions on sharing data publicly. For more information on unacceptable data access restrictions, please seehttp://journals.plos.org/plosone/s/data-availability#loc-unacceptable-data-access-restrictions.

Ethical restrictions apply here, as the data contains stories regarding the lives of residents, from the perspective of clients themselves, family and care professionals. Therefore, even if the names and other personally identifiable details were removed, the stories could still be linked to one of the residents. Because of the nature of the data, we have opted to not release it publicly. The data can still be inspected upon request through the AWO-L (s.aarts@maastrichtuniversity.nl).

The interview data won’t be shared publicly, but is available on request. However, the code and the models will become available on Zenodo and Github. 

5. Review Comments to the Author

Reviewer #1: Dear Authors,

Thank you for this possibility to peer review your interesting and important study paper. The study aims to understand the potential usefulness of text mining for qualitative research in long-term care for older adults. It compares text mining approach to manual approach regarding sentiment analysis and thematic content analysis in terms of accuracy, consistency, and expert feedback. This study is meaningful for the reasons which the authors state in their paper - manual analysis of large free-text-data-sets is time consuming and has risk of objectivity bias. Thus new effective analysis methods are needed.

Dear reviewer, thank you for your time and expertise in thoroughly reviewing our manuscript. Your detailed feedback have provided us with valuable insights and improved the comprehensibility of the manuscript. We appreciate the statement regarding the meaningfulness of our research, as we strive to explore efficient analysis methods that can overcome the challenges posed by manual analysis. We’ve made the necessary adjustments to manuscript to reflect your feedback.

These peer review comments are given from the point of view of non-native English user and nursing scientist who has used text-mining method in own research.

1. Is the manuscript technically sound, and do the data support the conclusions?

-Line 1: The tile might be stronger if the "interview data" would be stated in the title

The title has been adjusted to: “Comparing text mining and manual coding methods: analyzing interview data on quality of care in long-term care for older adults.”

-Line 38: In the abstract conclusions, I am not sure if you can say, based on your results, that text mining is potential for large data sets, as your sample was just couple of hundreds interview texts. I would prefer to conclude, that text mining is potential for analysis of free text data in the context of LTC for older adults. For future work you might suggest to test the method in large data sets (e.g. thousands-tens of thousands).

While our model was not tested on many thousands of samples, we can extrapolate from the literature that our model was based on [4]. Moreover, when conducting the analysis for this study, it took less than a minute to analyse the entire dataset on our hardware (an RTX 2060). We’ve added a paragraph to the discussion to clarify this: “Using deep learning models, such as those highlighted in this study, offers a distinct advantage in terms of speed. While deep learning models can process and analyse data within seconds, manual analysis, depending on the complexity and volume of the data, can span weeks to even months [41]. However, it's essential to recognize that the results from deep learning models might not always align perfectly with those of manual analysis. As such, researchers might find the need to fine-tune the outputs generated by text mining models. Despite this, the integration of deep learning significantly accelerates the qualitative analysis process, offering a more efficient alternative to traditional methods.” (p 19, l 362)

4. Delobelle, P., Winters, T., & Berendt, B. (2020). Robbert: a dutch roberta-based language model. arXiv preprint arXiv:2001.06286.

-The study presents results of original research and the sections are constructed according to journal's guidelines and are named and organized in logical order.

Thank you.

-The introduction section describes the judgement for the need of this study and states the aim of the study.

Thank you.

-Line 75: There should be reference for the gold standard of manual coding.

We have added a reference for this:

Song H, Tolochko P, Eberl JM, Eisele O, Greussing E, Heidenreich T, Lind F, Galyga S, Boomgaarden HG. In validations we trust? The impact of imperfect human annotations as a gold standard on the quality of validation of automated content analysis. Political Communication. 2020 Jul 3;37(4):550-72.

-Materials and Methods paragraph: You could add here your study design.

We have added the ‘study design’ heading.

-Lines 85-86: References for the two Text mining models should be presented.

Both the sentiment analysis as well as the thematic content analysis model were based on the same base model (i.e. RobBERT). We’ve added the appropriate references:

Delobelle, P., Winters, T., & Berendt, B. (2020). Robbert: a dutch roberta-based language model. arXiv preprint arXiv:2001.06286.

- Line 93 in Sample and Participants paragraph: A total of n=250 is confusing as earlier was said different n-numbers. In addition in Line 105 and Line 108 -> are presented different numbers. The numbers in different places needs more clarification for transparency.

The manuscript was adjusted to make this more clear. “A total of n = 250 interviews were conducted at five different LTC organizations in the southern part of the Netherlands. From those interviews, 234 were transcribed, 16 could not be transcribed due to poor audio quality. From the remaining 234 interviews, 61 were analysed manually using thematic content analysis. In addition, 103 interviews were analysed manually using sentiment analysis. All analysis in the manuscript were performed using those 61 and 103 interviews for the thematic content analysis and sentiment analysis respectively.” (p 7, l 104)

Should you describe also the experts' interviews in this section?

We have altered this part of the method section. "The data were coded by three research experts, each working in the Living Lab on Ageing and Long-Term Care for over 5 years. All these experts have a minimum of ten years of experience in conducting qualitative research." (p 8, l 126)

-Line 94: Who were interviewing the participants? How many interviewers?

To be more concise of who the interviews were, we have added this information. "A total of n = 35 interviewers conducted the interviews. These interviewers were part of the project 'Connecting Conversations,' which aims to assess the experienced quality of care in nursing homes from the resident’s perspective. They primarily come from a long-term care setting and have received specialized training to conduct these interviews. For a more comprehensive understanding of the 'Connecting Conversations' project, see Sion et. al. 2020a." (p 7,l 112) [2]. 

2. Has the statistical analysis been performed appropriately and rigorously?

-Line 113: Is the MAXQDA coding system free of charge or commercial software? Can you describe this?

MAXQDA is a commercial software, although free alternatives exist. 

-Line 121-123: Move this sentence to limitation section in discussion " Both models were capable of identifying where and how to code a text segment using a context of 512 words at most, which represents a technical limitation of the method [17,18]"

We have adjusted this in the limitations section: “Lastly, the analysis conducted in the current study only had context window of 512 words at most, which represents a technical limitation of the method [17,18]. This limits the textual context that the models have access to. These issues can be mitigated using large language models that are better able to capture the nuances and complexities of natural language (e.g. GPT-3) [25,37]. Such models can also handle a larger context of words. Whereas RobBERT has a maximum context length of 512, GPT-3 has a context of 4,096. However, such large language models cannot be used on most personal computers, as they require specialized hardware to run efficiently (i.e. GPUs or TPUs with large amounts of memory) [38]. Using these via online (cloud) systems could give rise to issues regarding the privacy of the interview participants. However, recent advances have shown that ‘smaller’ (i.e. more efficient) large language models can achieve similar results, and these models can be used on personal computers, unlike GPT-3 [39,40].” (p 20,l 388)

-Line 125: Who coded the sentiment analysis? One, two or more researchers?

Transcripts were coded by three researchers. This information was added to the manuscript.

3. Have the authors made all data underlying the findings in their manuscript fully available?

-Line 160: Could you present sample table of accuracy scores and confusion matrix?

The authors are unsure what the reviewer means with this specific comment. Confusion matrices are shown in figure 1 and 2. We have added additional statistics to the manuscript, including the average accuracy over all transcripts and their standard deviations. 

For the sentiment analysis we added the line: “The average accuracy over all transcripts was 88.7% with standard deviation of 8.6%. The minimum was accuracy was 52.1% and the maximum accuracy was 99.6%.” (p 13, l 227) and for the thematic content analysis: “The average accuracy over all transcripts was 81.9% with standard deviation of 8.5%. The minimum accuracy of any transcript was 43.1% and the maximum was 93.4%.” (p 14, l 256) We have elaborated on this in the discussion: “The variance of the accuracy between transcripts shows that a possible reason for lower accuracies could be due to factors that vary between transcripts , such as the quality of the transcription, the nature of the language used by the participants, or contextual factors that were not taken into account by the text mining or manual approach.” (p 18, l 335)

-Line 169-171: What was semantics consistency scale between very similar - very different?

We’ve adjusted the text to clarify this: “Semantic similarity is a value between 0% and 100%, where a higher percentage indicates that the results were more consistent.” In addition, we also included a reference [5]. (p 14, l 271).

5. Bölücü, Necva, Burcu Can, and Harun Artuner. "A Siamese neural network for learning semantically-informed sentence embeddings." Expert Systems with Applications, 214, 2023: 119103.

-Line 176-177: Research experts interview were not described in methods section.

As stated before, more information regarding the experts are provided in the methods section.

-Line 199-202: The sentence is unclear: "The text mining model was less accurate in determining what may be relevant to the organisation of a nursing home, as text which was not coded manually was often coded by text mining as either negative (4.3%) or positive (5.2%)."

This was clarified this by changing the sentence to: “Text mining often coded text as positive (4.3%) or negative (5.2%) that was not coded in the manual analysis”.

-Line 278: Discussion: The text is logical and clear in discussion, however I would suggest to add comparison of your results with previous studies. You have some references presenter, but not the comparison or mirroring your own results towards other studies.

Thank you for the statement regarding the clear discussion. For many studies, the model’s performance is compared to similar models on public datasets. However, in the current manuscript we’re working with a non-public dataset that has its own unique characteristics. This makes direct comparison with other studies impossible, as there are no established benchmarks or standards for our specific dataset. 

-Line 328...: Could you present some health care related example instead of the phrase ".... For example, given four sentences regarding a cat, a dog, a lion, and a wolf, it is ...."

The reviewer stated an important point here. Hence, this has been altered in the manuscript. “For example, given four sentences regarding a resident, a nurse, a resident's family member, and a visiting doctor, it is possible to split these sentences based on whether a person is a healthcare professional or not; however, it is also possible to split them based on whether a person is part of the nursing home staff.” (p 20, l 377)

-Line 359 in Conclusions section: Here also you use wording "large data sets" ... prefereably should not overestimate suitability for large data sets based on your results, just suitability for free text analysis in this context.

We have addressed this in an earlier comment and adjusted the discussion. We’ve added a paragraph to the discussion to clarify this: “Using deep learning models, such as those highlighted in this study, offers a distinct advantage in terms of speed. While deep learning models can process and analyse data within seconds, manual analysis, depending on the complexity and volume of the data, can span weeks to even months [3]. However, it's essential to recognize that the results from deep learning models might not always align perfectly with those of manual analysis. As such, researchers might find the need to fine-tune the outputs generated by text mining models. Despite this, the integration of deep learning significantly accelerates the qualitative analysis process, offering a more efficient alternative to traditional methods.” (p 19, l 362)

-References: Majority of the references are of high quality and from the last five years. However, 1/4 of the references are older than 10 years 

We appreciate your acknowledgment of the quality and recency of the majority of our references. Regarding the older references, it's crucial to highlight that, while currency in citations is often indicative of relevance in fast-evolving fields, foundational works can often remain pertinent long after their publication, as they form the basis of our models. In our selection of references, the older citations were included to provide historical context, foundational understanding, or to reference methodologies and theories that remain central to the topic even after a decade or more. We’ve added the following references:

1. Maycock, M. ‘I Do Not Appear to Have had Previous Letters’. The Potential and Pitfalls of Using a Qualitative Correspondence Method to Facilitate Insights Into Life in Prison During the Covid-19 Pandemic. International Journal of Qualitative Methods, 2021, 20: 16094069211047129.

2. Lee, P, Sebastien B, and Joseph P. "Benefits, limits, and risks of GPT-4 as an AI chatbot for medicine." New England Journal of Medicine, 388.13, 2023: 1233-1239.

and the peer-review of the reference is missing or unclear among the following references: Lines: 393, 400, 412, 417, 419, 428, 435, 447, 456, 459, 463, 465, 468, 472 and 475.

The references with "arXiv preprint arXiv" are submissions to a computer scientist database, but might not be yet peer reviewed or might have been even rejected from publication after submission. You should try to find the peer reviewed version of these articles and write references according to accepted/ published versions of the papers.

Many cutting-edge methods, particularly in the field of computer science, are initially published on platforms like arXiv. While these submissions might not have undergone the traditional scientific peer-review processes, they often come from reputable researchers or institutions and are accompanied by transparent resources such as source code and model weights. This open-access approach allows the broader community to inspect, validate, and build upon the work. We acknowledge the importance of peer-reviewed articles and always strive to reference them when available. However, given the rapid advancements in the field, we also believe in the value of these preprints as they represent the latest developments. Hence, we decided to keep the included references (as they are the foundation of our models) but also add some new, peer-reviewed references:

“43. Maycock, M. ‘I Do Not Appear to Have had Previous Letters’. The Potential and Pitfalls of Using a Qualitative Correspondence Method to Facilitate Insights Into Life in Prison During the Covid-19 Pandemic. International Journal of Qualitative Methods, 2021, 20: 16094069211047129.

44. Lee, P, Sebastien B, and Joseph P. "Benefits, limits, and risks of GPT-4 as an AI chatbot for medicine." New England Journal of Medicine, 388.13, 2023: 1233-1239.”

-Figure 1 and Figure 2: These figures do not open up for me. These needs revisions and clarifications. At the moment it looks to me, that Text mining and Manual analysis had exactly the same statistical values. I do not know how to read the Figures. In addition, you should understand the figures as stand-alone, without reading the manuscript text.

We’ve updated the caption of the figures to improve the clarify of the figures. These captions should allow the reader to understand the figures without reading the rest of the manuscript. (p 13, l 230)

4. Is the manuscript presented in an intelligible fashion and written in standard English? PLOS ONE does not copyedit accepted manuscripts, so the language in submitted articles must be clear, correct, and unambiguous. Any typographical or grammatical errors should be corrected at revision, so please note any specific errors here.

-As I am not native English user, I do not have comments for the English grammatic or typography.

I hope these comments help you to further develop your paper and to get it even stronger.

This paper deserves to get published, but needs some more details for transparency and in order to be able to use your method for replication.

Thank you for your constructive feedback. We believe the manuscript has improved greatly based on this feedback.

Reviewer #2: Summary Statement: This reviewer thanks the authors for their submission. Indeed, evaluation of information from clients in long-term care is relevant and methods to address their quality of care experience. More specifically, authors aim to evaluate if text mining can be accurate, consistent, and similar to expert review in both sentiment analysis and thematic content. The manuscripts is straightforward and quite readable but could be improved with further methodological and quantitative depth added as well as specific details in the discussion about how to improve the next study in the domain.

Strengths where no changes are required:

1) The methods of inclusion are well-described and include a robust number of participants (n=250) with written informed consent from 5 sites which seems adequate and appropriate for the evaluation.

2) The methods of themes into (14) key areas seemed appropriate and were well-described.

3) Evaluation of the text mining was performed using accuracy, consistency, and expert review which seemed appropriate.

We deeply appreciate your time and constructive feedback on our manuscript. We’ve carefully considered your comments and have adjusted the manuscript accordingly.

Weaknesses and areas of the manuscript that could be improved through further efforts:

1) The types of text mining could be further scientifically described in the introduction that are used later in the methods section:

Authors could better represent the current challenges with coding of qualitative data with clear description of methods and their performance characteristics. Recommend add 2-3 references in the introduction that specifically outline challenges in current methods with more specificity. Other authors have described with greater detail the text mining methods such as those described in the following (or could be alternative): Pranita Mahajan, Dipti P. Rana; International Journal of Innovative Technology and Exploring Engineering (IJITEE), ISSN: 2278-3075, Volume-9 Issue-2S, December 2019; or Annu Rev Biomed Data Sci. 2021 Jul 20;4:165-187. doi: 10.1146/annurev-biodatasci-030421-030931. Epub 2021 May 26. Additionally, further details about text mining models including InstructGPT and MM-CoT would be appropriate to cover in 1 sentence in the background/introduction.

Thank you for the insightful comments regarding the need for a more detailed discussion of text mining methods in the introduction section of our manuscript. We agree that clarifying the methods used and citing relevant challenges in the current methodologies can enrich the manuscript. Currently, Large Language Models (LLMs) are outperforming older rule-based and machine learning methods [6]. We have added the following to the introduction: “Recent advancements in the field of text mining have ushered in a variety of new techniques, each with its unique focus and application [34–36, 39, 46, 47]. Some models are particularly good at generating context-aware (e.g. InstructGPT), human-like text, while others excel at incorporating multi-modal data, such as text and images, for a more comprehensive analysis [34, 35, 36]. Moreover, there is a growing emphasis on adapting these models to run efficiently on consumer hardware [39]. Despite these strides in technology, there are still significant challenges in achieving the level of accuracy required for some tasks, and in many cases, human expertise continues to outperform automated methods [39].” (p 5, l 75)

6. Zhong Q, Ding L, Zhan Y, Qiao Y, Wen Y, Shen L, et al. Toward efficient language model pretraining and downstream adaptation via self-evolution: A case study on SuperGLUE. arXiv preprint arXiv:221201853. 2022.

2) While the authors used a Dutch language inclusive model, only a limited number of text word concepts were available (n=512 words). While this may not require a modification, authors in the discussion should further elaborate on the implications to the findings.

We have adjusted this in the limitations section: “Lastly, the analysis conducted in the current study only had context window of 512 words at most, which represents a technical limitation of the method [17,18]. This limits the textual context that the models have access to. These issues can be mitigated using large language models that are better able to capture the nuances and complexities of natural language (e.g. GPT-3) [25,37]. Such models can also handle a larger context of words. Whereas RobBERT has a maximum context length of 512, GPT-3 has a context of 4,096. However, such large language models cannot be used on most personal computers, as they require specialised hardware to run efficiently (i.e. GPUs or TPUs with large amounts of memory) [38]. Using these via online (cloud) systems could give rise to issues regarding the privacy of the interview participants. However, recent advances have shown that ‘smaller’ (i.e. more efficient) large language models can achieve similar results, and these models can be used on personal computers, unlike GPT-3 [39,40].” (p 20,l 388)

3) Expert feedback is inadequately described and should include a sentence referencing exactly in what capacity the individuals are considered expert. 

In We have altered this part of the method section. "The data were coded by three research experts, each working in the Living Lab on Ageing and Long-Term Care for over 5 years. All these experts have a minimum of ten years of experience in conducting qualitative research." (p 8, l 126)

The statistical methods utilized in the comparison should also be briefly presented in the methods.

We added the text: “Accuracy was calculated using the formula: (TP + TN) / (TP + TN + FP + FN). In this case, TP is the true positive (i.e. where a code is present in both analyses), TN is the true negative (i.e. where a code is absent in both analyses), and FP is the false positive (i.e. where a code is predicted to be present but is absent in the manual analysis), while FN is the false negative (i.e. where a code is predicted to be absent but is present in the manual analysis). These components help us assess the accuracy of the model's predictions and its performance overall.” (p 10, l 184) A reference was included:

Zhou Z-H. Machine Learning. Springer Nature 2021. Available from: https://doi.org/10.1007/978-981-15-1967-3 [Accessed April 12, 2023].

For the consistency analysis we’ve added references to relevant literature the use vector embeddings to check similarity:

Bölücü, Necva, Burcu Can, and Harun Artuner. "A Siamese neural network for learning semantically-informed sentence embeddings." Expert Systems with Applications, 214, 2023: 119103.

For the expert feedback, we didn’t have any quantitative results to apply statistics to.

4) Authors describe the difficulties inherent in the analyses with discrepancies which can also be due to as well the multiple themes that are present but could expand upon how this can be mitigated in the discussion.

The reviewer stated an interesting and important point here. The problem with multiple themes is something very difficult to mitigate for the current dataset, as the themes are all highly correlated. In the ‘Limitations’ section we suggest that larger models that have been trained for longer are more capable of distinguishing between the nuances in the text. Other possibilities to mitigate the issue, are using more data or creating themes are split more clearly. Data quality therefore also plays are role. 

5) Overall, the manuscript could be improved through the additional enrichment of quantitative findings as well as depth in the methodological approach.

We’ve clarified the statistical methods in the ‘methods’ section. Moreover, we’ve added the average and standard deviation of the accuracy scores over all transcripts.

For the sentiment analysis we added the line: “The average accuracy over all transcripts was 88.7% with standard deviation of 8.6%. The minimum was accuracy was 52.1% and the maximum accuracy was 99.6%.” (p 13, l 227) and for the thematic content analysis: “The average accuracy over all transcripts was 81.9% with standard deviation of 8.5%. The minimum accuracy of any transcript was 43.1% and the maximum was 93.4%.” (p 14, l 256) We have elaborated on this in the discussion: “The variance of the accuracy between transcripts shows that a possible reason for lower accuracies could be due to factors that vary between transcripts , such as the quality of the transcription, the nature of the language used by the participants, or contextual factors that were not taken into account by the text mining or manual approach.” (p 18, l 335)

Minor Editing Recommendations:

1) Correction recommendation: In the introduction, there is a period that needs replaced by a comma, line 48:

To be able to analyze these data, 48 researchers often conduct a so-called coding analysis [2,5]. which involves manually.

Thank you for your thorough review. We’ve altered this.

References

1. Song H, Tolochko P, Eberl JM, Eisele O, Greussing E, Heidenreich T, Lind F, Galyga S, Boomgaarden HG. In validations we trust? The impact of imperfect human annotations as a gold standard on the quality of validation of automated content analysis. Political Communication. 2020 Jul 3;37(4):550-72.

2. Sion K, Verbeek H, de Vries E, Zwakhalen S, Odekerken-Schröder G, Schols J, Hamers J. The Feasibility of Connecting Conversations: A Narrative Method to Assess Experienced Quality of Care in Nursing Homes from the Resident's Perspective. Int J Environ Res Public Health. 2020 Jul 15;17(14):5118. doi: 10.3390/ijerph17145118. PMID: 32679869; PMCID: PMC7400298.

3. Wang H. Efficient algorithms and hardware for natural language processing (Doctoral dissertation, Massachusetts Institute of Technology). 2020.

4. Delobelle, P., Winters, T., & Berendt, B. (2020). Robbert: a dutch roberta-based language model. arXiv preprint arXiv:2001.06286.

5. Bölücü, Necva, Burcu Can, and Harun Artuner. "A Siamese neural network for learning semantically-informed sentence embeddings." Expert Systems with Applications 214 (2023): 119103.

6. Zhong Q, Ding L, Zhan Y, Qiao Y, Wen Y, Shen L, et al. Toward efficient language model pretraining and downstream adaptation via self-evolution: A case study on SuperGLUE. arXiv preprint arXiv:221201853. 2022.

---

## [Decision Letter · Decision Letter 1]

25 Sep 2023

Comparing text mining and manual coding methods: analysing interview data on quality of care in long-term care for older adults

PONE-D-23-11158R1

Dear Dr. Hacking,

We’re pleased to inform you that your manuscript has been judged scientifically suitable for publication and will be formally accepted for publication once it meets all outstanding technical requirements.

Kind regards,

Baby Gobin

Academic Editor

PLOS ONE

Additional Editor Comments (optional):

Reviewers' comments:

Reviewer's Responses to Questions

**Comments to the Author**

1. If the authors have adequately addressed your comments raised in a previous round of review and you feel that this manuscript is now acceptable for publication, you may indicate that here to bypass the “Comments to the Author” section, enter your conflict of interest statement in the “Confidential to Editor” section, and submit your "Accept" recommendation.

Reviewer #1: All comments have been addressed

Reviewer #2: All comments have been addressed

2. Is the manuscript technically sound, and do the data support the conclusions?

Reviewer #1: Yes

Reviewer #2: Yes

3. Has the statistical analysis been performed appropriately and rigorously? 

Reviewer #1: Yes

Reviewer #2: Yes

4. Have the authors made all data underlying the findings in their manuscript fully available?

Reviewer #1: Yes

Reviewer #2: Yes

5. Is the manuscript presented in an intelligible fashion and written in standard English?

Reviewer #1: Yes

Reviewer #2: Yes

6. Review Comments to the Author

Reviewer #1: Dear Authors,

Thank you for your thorough and detailed responses to all of the presented questions and suggestions. You have improved your article stronger. Also your reasoning for using The references with "arXiv preprint arXi " is acceptable. I am delighted to have the possibility to review this excellent scientific paper advancing evidence about text mining possibilities in health sciences research. I wish you interesting research projects in the future and I am looking forward to reading your further publications!

Reviewer #2: Summary Statement: This reviewer thanks the authors for their revision on the submission. The authors have point by point addressed each reviewer concern with careful and thoughtful attention to each detail. Specifically, they have improved the description of the methodological and quantitative approach as well as the discussion about how to improve the next study in the domain. As a result, the overall manuscript reads much better and contributes to the knowledge in this important domain.

(No Change in Comments) Strengths where no changes are required:

1) The methods of inclusion are well-described and include a robust number of participants (n=250) with written informed consent from 5 sites which seems adequate and appropriate for the evaluation.

2) The methods of themes into (14) key areas seemed appropriate and were well-described.

3) Evaluation of the text mining was performed using accuracy, consistency, and expert review which seemed appropriate.

Weaknesses and areas of the manuscript that could be improved through further efforts were addressed to the degree appropriate in the revision:

1) The types of text mining could be further scientifically described in the introduction that are used later in the methods section:

Authors have added the recommended references in the introduction that specifically outline challenges in current methods with more specificity.

2) While the authors used a Dutch language inclusive model, only a limited number of text word concepts were available (n=512 words). Authors have adequately elaborated on the implications to the findings.

3) Authors have added further details related to the expert feedback and their capacity to do so.

4) Authors have further expanded upon the discrepancies which can be mitigated in the discussion.

5) Authors have further enriched the quantitative findings as well as depth in the methodological approach.

Minor Editing Recommendations:

Authors have made the appropriate minor editing changes, and no further changes to recommend.

7. PLOS authors have the option to publish the peer review history of their article (what does this mean?). If published, this will include your full peer review and any attached files.

Reviewer #1: No

Reviewer #2: **Yes: **Amy M Sitapati, MD

---

## [Editor Report · Acceptance letter]

23 Oct 2023

PONE-D-23-11158R1 

Comparing text mining and manual coding methods: analysing interview data on quality of care in long-term care for older adults 

Dear Dr. Hacking:

I'm pleased to inform you that your manuscript has been deemed suitable for publication in PLOS ONE. Congratulations! Your manuscript is now with our production department. 

Kind regards, 

on behalf of

Dr. Baby Gobin 

Academic Editor

PLOS ONE